# Design of Fairway Width Based on a Grounding and Collision Risk Model in the South Coast of Korean Waterways

Won-Sik Kang [1], Young-Soo Park [2,*], Myoung-Ki Lee [3] and Sangwon Park [4]

1 Korea Maritime Transportation Safety Authority, Sejong 30100, Korea; wskang84@komsa.or.kr
2 Division of Navigation Convergence Studies, Korea Maritime and Ocean University, Busan 49112, Korea
3 Research Institute of Maritime Industry, Korea Maritime and Ocean University, Busan 49112, Korea; lmk0620@g.kmou.ac.kr
4 Korea Maritime Institute, Busan 49111, Korea; psw6745@kmi.re.kr
* Correspondence: youngsoo@kmou.ac.kr; Tel.: +82-51-410-5085

**Abstract:** As a method of reviewing the design of new fairways and the redesign of existing fairways, we proposed a fairway design plan based on the collision and grounding probability, considering vessel traffic. A case study was conducted on the four traffic separation schemes (TSS) on the southern coast, which is the most complex coast among Korean coastal waters. The evaluation items of Korea's Maritime Traffic Safety Assessment Scheme, the PIANC Guide, and the Port and Fishing Port Design Standards were all satisfied; however, some fairways had very high ship congestion at specific times, exceeding the fairway capacity. For each target fairway, the collision risk using the environmental stress (ES) model and the grounding risk using the IWRAP Mk II Model were analyzed. The grounding risk was found to be equally good, but the aggregation environmental stress ($ES_A$) value, according to the ES model, was high in three fairways. The widths of the three fairways with high risk were partially expanded, and, thus, were re-evaluated. The overall $ES_A$ was reduced, and the psychological burden of operators due to the marine environment, such as the fairways, was significantly eased. Based on the results of this study, it would be beneficial to apply a design scheme using collision and stranded risk models when designing new fairways or reviewing existing fairways. An appropriate fairway design plan is prepared that, through further research using various evaluation models and techniques, could be useful in coastal waters in the future.

**Keywords:** fairway design; environment stress model (ES); IWRAP Mk II; traffic congestion analysis; traffic separation scheme (TSS)

## 1. Introduction

In preparation for the global warming caused by the excessive use of fossil fuels, countries worldwide are currently establishing and implementing policies to increase renewable energy sources. According to the 3020 implementation plan, Korea is planning to increase the proportion of renewable energy generation from the current 7% to 20% by 2030. In particular, at sea, it aims to expand the scale of offshore wind power to 12 GW by 2030 [1]. However, because of the nature of offshore wind power, the water depth is restricted; therefore, when offshore wind power is installed in a narrow sea area, side effects may occur, such as the infringement of existing shipping fairways and a deterioration of safety [2–5]. Governments around the world are establishing various policies to secure their traffic safety and strengthen safety systems [6–8]. As a response to this situation, the Ministry of Oceans and Fisheries has established and promoted a national integrated maritime transportation network plan to establish a safe and efficient maritime transportation system for the entire coastal waters of South Korea [9,10].

Research related to fairway design has been conducted in various ways in the past, and recently, research related to autonomous navigation or AI operation has been conducted [11,12]. Paulauskas (2013) studied the width of a channel in a port, and Gucma et al. (2020)

conducted a study on the width of a safe channel in a curved channel. Kim and Lee (2020) studied the minimum route width by ship type through double speed simulation [13–15].

In 1985, the International Maritime Organization (IMO) adopted Resolution A. 572(14) of the General Provision Resolution on the route of ships. This Resolution was adopted to improve the navigational safety of ships in areas with high sea traffic density or limited water depth, and defines the Routing System, Traffic Separation Scheme (TSS), and Separation Zone. Although the purpose of installing the route is to protect the vessel from a high density of marine traffic and low water areas, that is, to prevent collisions and grounding accidents of the vessel, there is a tendency to design it with a focus on the collision of the vessel only. Korea's Maritime Safety Act, adopting COLREG Rule 10 in Article 68, proposes to install a TSS in areas with a high risk of collision due to high traffic density. The Korea Maritime Safety Tribunal classifies marine accidents into 14 categories, such as collision, capsize, sinking, fire, explosion, and grounding [16]. Yildiz, S. et al. (2021) conducted a study on the effects of collisions and stranding caused by inexperience or poor use of navigational equipment by ship operators [17,18]. The main purpose of installing the route is to prevent collisions and grounding accidents, so it is necessary to design the route in consideration of this. As such, previous studies have mainly been conducted using advanced technology or related studies on port access waters. PIANC or the Port and Fishing Port Design Standards also numerically proposed the width of the fairway; however, the standards simply present the minimum design based on the maximum number of ships passing the target sea area, and the safety of the vessel may vary significantly based on the sea area characteristics and volume.

In contrast, as vessels become larger and faster, the appropriateness of the previously designed fairways must be periodically checked. The Maritime Traffic Safety Assessment Scheme (MTSA) in Article 15 of the Maritime Safety Act, which is a representative means of assessment, requires operators who want to implement maritime development projects to prepare safety measures through professional evaluation in advance. The MTSA designates essential and optional evaluation items, depending on the project type. To designate a new fairway or change an existing fairway, marine traffic surveys and vessel control simulation evaluations must be performed, while marine traffic congestion evaluations and marine traffic simulation evaluations are optional, according to Article 12 of Maritime Traffic Safety Assessment Implementation Guidelines. The MTSA has been implemented for more than 10 years since the introduction of the system (January 2010), and it has been reported that it is effective in various aspects. However, the MTSA also does not consider expanding or adjusting the scope of the fairway if it meets the basic design criteria, such as the existing PIANC or Port and Fishing Port Design Standards. If the fairway is evaluated as being congested through safety evaluations using simulations and the collection of opinions, safety measures are established to reduce periods with high congestion using VTS.

The design of the fairway should consider both safety and economic feasibility, and check whether the vessel is safe from the risks of collision and grounding, in accordance with changes in the marine traffic environment. For this purpose, in this study, a previously established fairway evaluation was conducted, using a model to evaluate the collision and grounding probability of a vessel, and the feasibility of the evaluation according to environmental changes was reviewed [19]. Figure 1 shows a flow chart of this study.

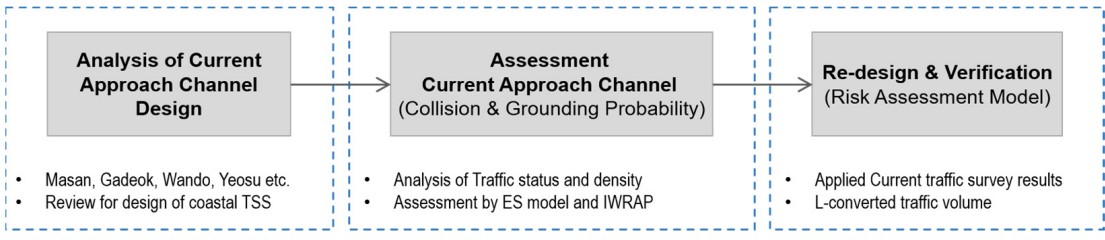

**Figure 1.** Flowchart of the study.

Based on the collision and grounding probability model, a case study was performed by selecting the existing fairway, to examine whether the safety of the vessel's passage can be sufficiently secured. In particular, a coastal TSS was selected as an evaluation target by targeting the southern coast, which exhibits various complex traffic flows along the coast of Korea [20]. In this study, the general status and maritime traffic status of Masan, Gadeok, Wando fairway, and Yeosu-specific sea fairway were selected for evaluation and investigated. Using this status, PIANC—which is the basis for the design of the existing fairways—and the Port and Fishing Port Design Standards were analyzed, and whether they were satisfied was reviewed. Maritime traffic congestion, one of the main evaluation items in the MTSA, was evaluated and used as basic data for the analysis of fairway adequacy. Consequently, the adequacy was evaluated by applying a collision and grounding probability model for the target fairway, redesigning the width of the fairway for some fairways with high collision risk, and then, re-evaluating the collision risk evaluation model for the redesigned fairway.

## 2. Status Analysis of the Fairway

### 2.1. Selection of Fairways to Be Analyzed

Korea is a peninsula country surrounded by seas to the east, west, and south. The main traffic flow moves from the West Sea to the East Sea through the South Sea, and a TSS is installed in the middle of the main flow to ensure smooth traffic flow. There are many islands in the West and South Seas, and among them, the southern coast has an area and a depth of water that allows ships to pass between the islands [21].

Figure 2 shows the traffic flow formed on the southern coast of Korea.

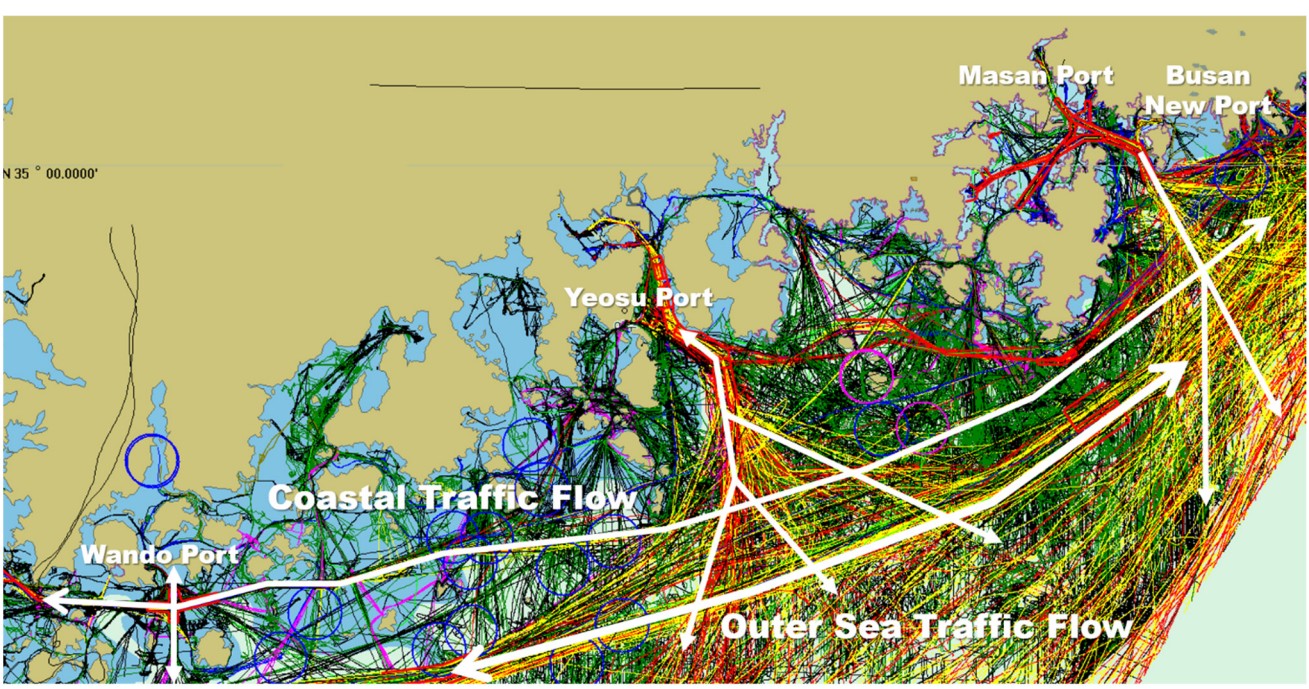

**Figure 2.** Major traffic flows along the southern coast of Korea.

In the middle of the coastal waters, Hoenggansudo Recommended Fairway, Wando Fairway TSS, Sodeokudo Recommended Fairway, and Samcheonpo Port are installed, to form major traffic flows from the east to west. In addition, ships enter and leave large-scale trade ports, such as the Wando Port, Yeosu Port, Gwangyang Port, Hadong Port, Samcheonpo Port, Tongyeong Port, Okpo Port, Busan New Port, Masan Port, Jinhae Port, and Busan Port; thus, there is a complex traffic flow, which naturally crosses the east and west.

In this study, the appropriateness of the existing installed fairway width was reviewed, considering the marine traffic volume and traffic environment. As mentioned above, we targeted the southern coastal waters, where complex traffic flows between the east and west and the north and south are formed. In addition, the official fairway, whose width and length are known, was selected to conduct an evaluation. Regarding fairways within a port, because the speed of the ship is restricted or the ship is controlled through VTS control, only fairways outside a port and allowing free passage flow without external interference were selected. Figure 3 shows the target fairway of this study.

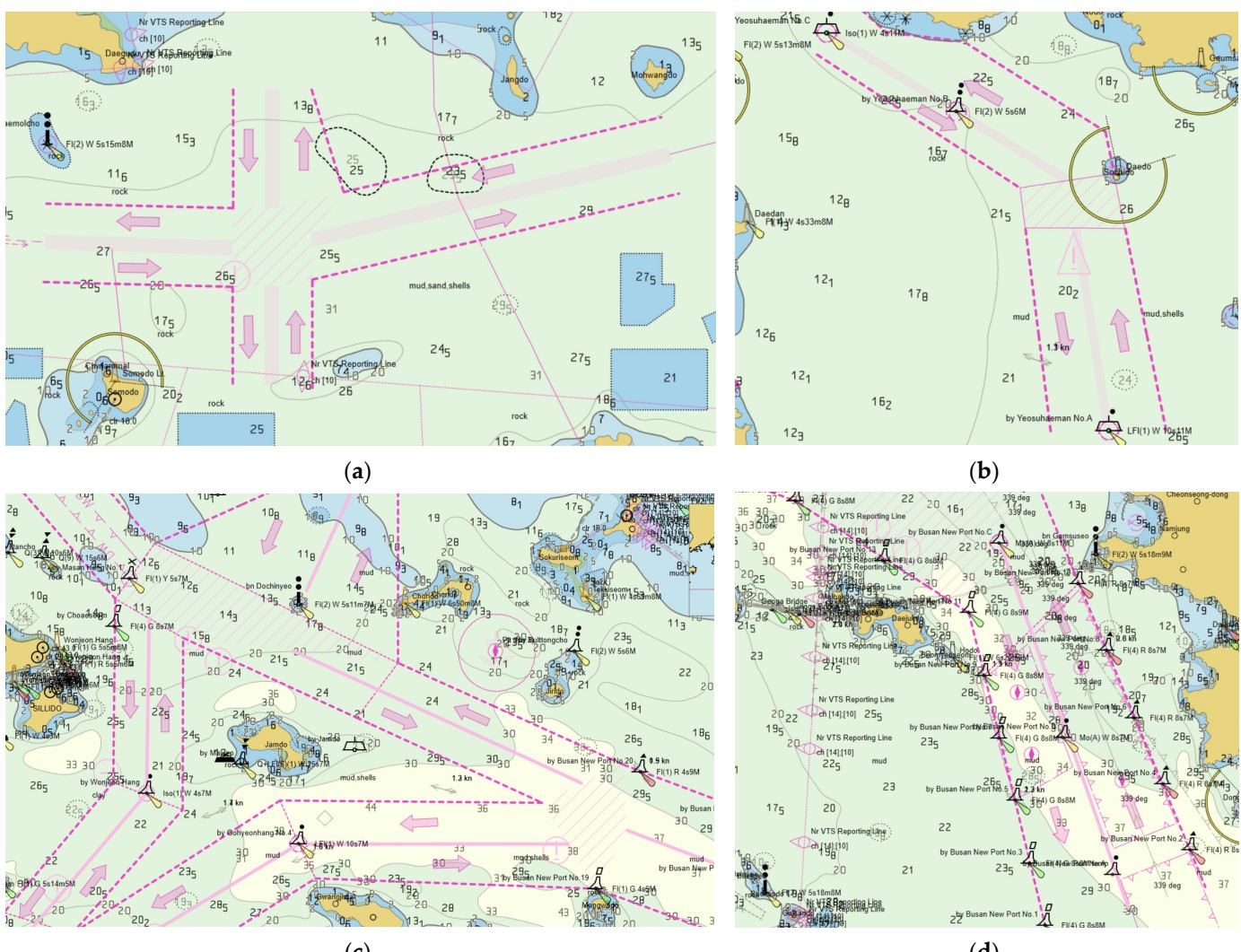

**Figure 3.** (**a**) Wando Port access and crossing fairways near the Wando Port, (**b**) Yeosu Area fairways in traffic safety-specific waters, (**c**) Masan Fairway connecting Masan Port, Jinhae Port, and Tongyeong Port, and (**d**) Gadeok Fairway Near Busan New Port.

The official out-of-port fairways formed on the southern coast of the South Sea are the Wando fairway near Wando Port, the fairway within the Yeosu area in specific traffic safety waters near Yeosu Port and Gwangyang Port, the fairway near the Masan Port, and the Gadeok fairway near Busan New Port. In this study, we selected the relevant fairway as the target fairway, the appropriateness of the existing fairway was reviewed, and the adequacy of the improved fairway was compared and reviewed based on the analysis results through re-adjustment and evaluation.

*2.2. General Status Analysis of Fairways*

Through Article 31 of the Maritime Safety Act, the Minister of Oceans and Fisheries designates an official fairway as an area recognized as likely to cause a marine accident due to natural conditions such as topography and current, or ship traffic in the water area through which ships pass, and the matters necessary for ship navigation safety. Except for port areas, the fairways officially existing along the Korean coast are divided into (1) three fairways within a specific area for traffic safety, (2) three fairways to which the traffic separation method is applied, and (3) 28 fairways announced by the Regional Office of Oceans and Fisheries. Of the total 34 official coastal fairways, the 21 fairways along the southern coast account for more than half. Of the 21 coastal fairways in the South Sea, the width of seven fairways can be clearly measured, and in this study, the remaining four official fairways were selected, except for the three fairways using the traffic separation method and located outside the main islands on the Southern Sea coast [20,21].

Table 1 shows the general status of the Wando Port access and crossing fairways, the Yeosu fairway in the specific traffic safety waters, the Masan fairway, and the Gadeok fairway.

**Table 1.** General status of research fairways.

| TSS Name | Width (m) | Width (m) by Bound | | Depth (m) |
|---|---|---|---|---|
| Wando Port Entry and Crossing | 1660 | West Bound | 680 | 12.6 |
| | | East Bound | 680 | |
| | | Separation Zone | 300 | |
| Traffic Safety Specific Water 'Yeosu Area' | 2000 | West Bound | 1120 | 20.2 |
| | | East Bound | 870 | |
| | | Separation Zone | 10 | |
| Masan | 1660 | West Bound | 800 | 21.0 |
| | | East Bound | 850 | |
| | | Separation Zone | 10 | |
| Gadeok-sudo | 1680 | North Bound | 900 | 20.3 |
| | | South Bound | 770 | |
| | | Separation Zone | 10 | |

Wando Port entry and crossing fairways were established based on Article 31 of the Maritime Safety Act (designation of fairways). This is an area where fishing is prohibited, and installation of fishing gear, such as fishing nets, and dumping are prohibited. Ships sailing from east to west and ships passing north and south for entry and departure from Wando Port intersect here. The width of the fairway is the same, at 680 m in one direction, and the separation zone is formed in the same way at 300 m. The lowest water depth in the fairway is ~12.6 m, and the longest section of the fairway from east to west is ~13.5 km.

The Yeosu Area on Traffic Safety Specific Waters was established based on Article 10 of the Maritime Safety Act (establishment of traffic safety specific sea area, etc.). Ship traffic control is enforced on the fairway, aquaculture is restricted, and prior permission is required for construction and work to be performed in the relevant sea area. It is a sea area where vessel navigation and maritime traffic are mainly managed; in particular, there is a reef on the right side of the middle of the fairway, so ships sailing to the North must avoid it. The lowest water depth in the fairway is ~20.2 m, and the longest section of the fairway is ~18.9 km.

The Masan Fairway was established based on Article 31 (Designation of Fairways) of the Maritime Safety Act. This is also an area in which fishing is prohibited, and installation of fishing gear, such as fishing nets, and dumping are prohibited. The target sea area is a fairway that serves as a means of entry into the large trade ports such as Masan Port, Jinhae Port, Tongyeong Port, and Gohyeon Port, and many ships pass through. The width of the fairway is 800 m on the West Bound and 850 m on the East Bound, and the lowest water depth in the fairway is ~21.0 m.

The Gadeok fairway was established based on Article 31 of the Maritime Safety Act (designation of fairways). This is also an area where ships are left unattended, and installation of fishing gear, such as fishing nets, and dumping are prohibited. The fairway is used to enter and depart Busan New Port, and traffic is active. The width of the fairway is 900 m on the North Bound and 770 m on the South Bound, and the lowest water depth in the fairway is ~20.3 m.

### 2.3. Analysis of Appropriate Fairway Width Based on Design Standards

To respond to the large-scale marine accidents that occurred continuously in Northern Europe in the 1960s, a study on the traffic separation method of ships was initiated in Europe (Ministry of Oceans and Fisheries, 2005). Afterwards, in 1968, the International Maritime Organization (IMO) developed a recommendation on the traffic separation method for 41 water zones, and after this recommendation, the number of participating waters significantly increased [22].

The effectiveness of the traffic separation method has been presented in various studies. A study on the Strait of Dover in the UK showed that after the installation of the traffic separator, marine accidents decreased by approximately 23% compared to before installation [23,24]. According to Park et al. (2003), a risk analysis using the comprehensive environmental stress model indicated that the ship operator's burden on operating was reduced by approximately 32% compared to the previous status after the installation of the traffic separation zone, and it was observed that marine accidents were reduced [25].

Although the IMO states that the fairway width of the traffic separation method should consider the traffic density, traffic mode, and available ship operating waters, it does not provide specific numerical values.

In maritime traffic safety diagnosis, the Port and Fishing Port Design Standards and the Harbor Approach Channels Design Guidelines of the International Association of Water Transportation Facilities (PIANC Rule) are reviewed to determine the appropriateness of the design of fairways. In these design standards, when designing the width of a channel approaching a port, based on the size of the largest vessel passing through the target sea, one-way and two-way traffic are separated and presented as standards [26–28].

In the PIANC Guideline, the width of the fairway is determined according to the characteristics of the ship, sea environment, navigation speed, shape of the seabed, and type of cargo [29].

Equation (1) shows the required path width of a one-way channel according to the PIANC Guideline:

$$W = W_{BM} + \sum W_i + W_{BR} + W_{BG} = W_M + W_{BR} + W_{BG} \tag{1}$$

Equation (2) shows the required width of a two-way channel according to the PIANC Guideline:

$$W = 2W_{BM} + 2\sum W_i + W_{BR} + W_{BG} + \sum W_P = 2W_M + W_{BR} + \sum W_P + W_{BG} \tag{2}$$

where

| | |
|---|---|
| $W_{BM}$ | Basic maneuvering lane; |
| $\sum W_i$ | Additional widths to allow for the effects of wind, etc.; |
| $W_{BR}$, $W_{BG}$ | Bank clearance; |
| $\sum W_P$ | Passing distance between both maneuvering lanes $W_M$. |

Table 2 shows the effects of ship speed, sea environment, navigation method, and the range of basic and additional fairway width conditions suggested in the PIANC Guide.

**Table 2.** Basic maneuvering lane and additional channel width of PIANC.

| Width | | Basic or Additional Width Range |
|---|---|---|
| Basic Maneuvering Lane $W_{BM}$ | | 1.3~1.8 B * |
| Additional Factor | Vessel Speed | 0~0.1 B |
| | Wind | 0.1~1.1 B |
| | Current | 0~1.6 B |
| | Wave | 0~1.0 B |
| | Navigation Aids | 0~0.4 B |
| | Bottom surface | 0~0.2 B |
| | Depth | 0~0.4 B |
| Two-Way Traffic | | 1.0~2.0 B |
| Bank Clearance | | 0~1.3 B |

\* B: Breadth of Target Vessel.

Based on Equations (1) and (2), when the conditions mentioned in Table 2 are applied, the appropriate bidirectional fairway width according to the PIANC Guide is analyzed to be a minimum of 3.8 B and a maximum of 18.6 B.

Similarly, in Korea's Port and Fishing Port Design Standards, the fairway width should be determined by fully considering conditions such as the specifications and characteristics of the target vessel, marine environment, and traffic situations. However, considering the traffic situation, clear standards are presented to a certain extent. Where two-way traffic is not expected, a fairway width of 0.5 L or more is preferred, and where two-way traffic is expected, the default fairway width is 1.0 L or more. In the case of a long fairway, a fairway width of 2 L or more must be secured. Table 3 shows the fairway width standards according to the port and fishing port design standards [30].

**Table 3.** Criteria for fairway width according to the Port and Fishing Port Design Standards.

| Type | Width Range |
|---|---|
| One-way | 0.5~1.0 L * |
| Two-way | 1.0~2.0 L |

\* L: Length of Target Vessel.

## 3. Conformity Review According to Fairway Design Standard

As previously analyzed, the criteria for determining the appropriate fairway width in the process of designating a new fairway or inspecting an existing fairway in coastal waters have not been clearly presented. However, the standard for designing the minimum fairway width based on the maximum sailing vessel is presented in the PIANC Guide and the Port and Fishing Port Design Standards. In this section, we investigated the largest sailing vessel for the four fairways selected, and reviewed whether it met the appropriate fairway width, according to the PIANC Guide and port and fishing port design standards. To analyze the appropriateness through an analysis of the fairway width and maritime traffic volume, which was simply reviewed based on the largest transit vessel, the maritime traffic volume was investigated and a maritime traffic congestion assessment, which is the main evaluation item in the maritime traffic safety diagnosis system, was conducted.

### 3.1. Analysis of Vessel Traffic Status

To review the adequacy of the design of the PIANC Guide for the South Sea coast fairway and the design criteria for ports and fishing ports, the current status of the maritime traffic was analyzed. Figure 4 shows the traffic flow for each target fairway. Although using data of one year for the maritime traffic analysis period is the preferred method, the traffic flow in the target sea was analyzed for 7 days, from 24 May 2020 to 30 May 2020, referring to the results of previous studies [31].

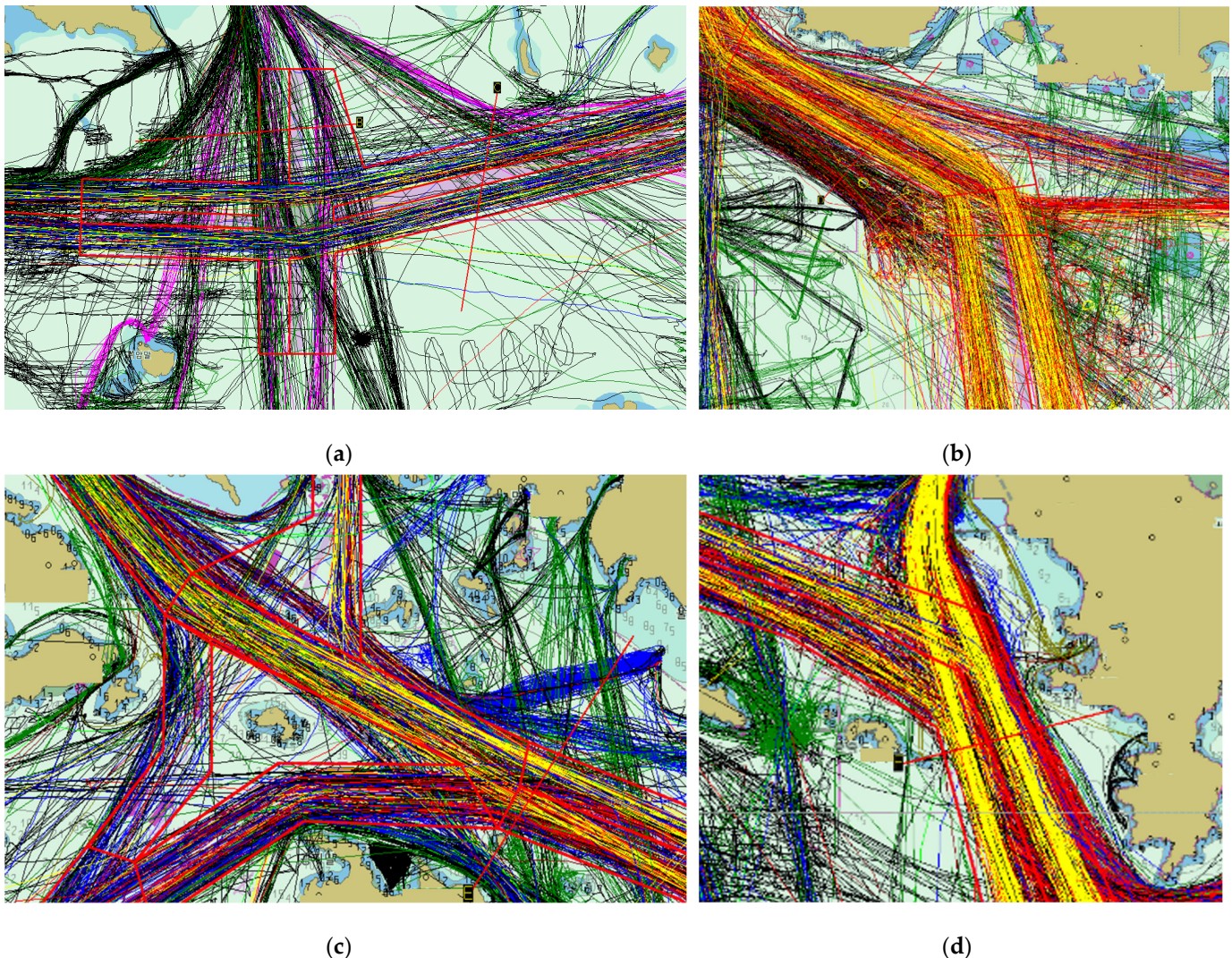

**(a)**                                                                 **(b)**

**(c)**                                                                 **(d)**

**Figure 4.** (**a**) Maritime traffic flow in the entrance and crossing fairways near Wando Port, (**b**) maritime traffic flow in the Yeosu area in specific traffic safety waters, (**c**) maritime traffic flow in the Masan fairway, and (**d**) maritime traffic flow in the Gadeok fairway.

As a result of analyzing the traffic flow of each fairway, we found that most ships navigate along the fairway; however, small- and medium-sized ships that are not considerably restricted by water depth may deviate slightly from the fairway. However, because this does not deviate from the main traffic flow, the research was conducted focusing on the main traffic flow. Table 4 shows the detailed maritime traffic analysis data for each fairway.

A total of 779 ships sailed through the Wando fairway over 7 days; furthermore, an average of 4.6 ships passed per hour. The largest vessel using the fairway was a 161 m long vessel, and the vessels passed in the following order: fishing vessels (13.9%), passenger vessels (13.2%), towing vessels (8.6%), cargo vessels (5.8%), and dangerous goods carriers (3.5%).

A total of 1274 ships passed through the fairway within the Yeosu area in the traffic safety specific sea area, and approximately 7.6 ships passed per hour. The largest vessel using this fairway was 400 m long, while ships such as dangerous goods carriers (35.1%) and cargo ships (22.3) mainly passed.

**Table 4.** Results of maritime traffic survey by target fairway.

| Category | | Wando | Yeosu | Masan | Gadeok |
|---|---|---|---|---|---|
| Total Number of Vessels | | 779 | 1274 | 957 | 1002 |
| Average Number of Vessels per day | | 111.3 | 182.0 | 136.7 | 143.1 |
| Average Number of Vessels per hour | | 4.6 | 7.6 | 5.7 | 6.0 |
| Max. Number of Vessels per hour | | 17 | 20 | 15 | 18 |
| Length of Largest Vessel (m) | | 161 | 400 | 345 | 400 |
| Number and Proportion by Vessel Type | Tanker | 27 (3.5%) | 447 (35.1%) | 96 (10.0%) | 193 (19.3%) |
| | Cargo ship | 45 (5.8%) | 284 (22.3%) | 87 (9.1%) | 301 (30.0%) |
| | Passenger | 103 (13.2%) | 13 (1.0%) | - | - |
| | Towing Vessel | 67 (8.6%) | 32 (2.5%) | 275 (28.7%) | 80 (8.0%) |
| | Fishing Vessel | 108 (13.9%) | 35 (2.7%) | 119 (12.4%) | 14 (1.4%) |
| | ETC | 429 (55.0%) | 463 (36.4%) | 380 (39.8%) | 414 (41.3%) |
| Number and Proportion by Vessel Length | ~50 m | 661 (84.9%) | 642 (50.4%) | 784 (81.9%) | 588 (58.7%) |
| | 51~100 m | 62 (8.0%) | 273 (21.4%) | 103 (10.9%) | 125 (12.4%) |
| | 101~150 m | 30 (3.9%) | 152 (11.9%) | 32 (3.3%) | 83 (8.3%) |
| | 151~200 m | 26 (3.2%) | 124 (9.7%) | 28 (2.9%) | 50 (5.0%) |
| | 201~250 m | - | 27 (2.1%) | - | 5 (0.5%) |
| | 251~300 m | - | 24 (1.9%) | 7 (0.7%) | 59 (5.9%) |
| | 301~350 m | - | 26 (2.0%) | 3 (0.3%) | 58 (5.8%) |
| | 351 m~ | - | 6 (0.6%) | - | 34 (3.4%) |

A total of 957 ships passed through the Masan fairway over 7 days, and it was analyzed that approximately 5.7 ships passed per hour. The largest vessel using the fairway was 345 m long, and towing vessels (28.7%), fishing vessels (12.4%), dangerous goods carriers (10.0%), and cargo ships (9.1%) passed in the given order.

A total of 1002 ships passed through the Gadeok fairway over 7 days, with approximately 6.0 ships passing per hour. The largest vessel using the fairway was 400 m long, and cargo ships (30.0%), dangerous goods carriers (19.3%), and towing vessels (8.0%) passed in the given order.

*3.2. L-Converted Traffic Volume*

Each sea area has different characteristics of passing ships, based on the existence of a water area near the port and the characteristics of the natural environment, such as water area [32]. Considering this, in this study, the concept of L-converted vertebrae was introduced and measured for ships navigating the sea route and not for simple ships. In marine traffic analysis, treating a small vessel of 10 m in length and a vessel of 400 m in length as one traffic volume is inappropriate. Even for a single ship, the size of the sea area occupied by the length and scale of the ship is different, and the degree of risk to the surroundings and the size of the sea area required for safe navigation are different. Converted traffic volume is a quantified relationship that considers the size of the ship, and when converted based on the length of the ship and when converting the number of ships, it is called the L-converted traffic volume. In general, when calculating the L conversion factor, the standard ship length is set as the standard, and the conversion factor is taken and integrated. We set the length of the standard vessel to 82 m, and the amount of traffic was converted [29].

Figure 5 shows the L-converted traffic volume, converted based on the length of all ships passing the target fairway by time zone for 7 days.

The maximum L-converted traffic volumes by time of the Wando fairway, the Yeosu area in a Specific Traffic Safety Waters, the Masan fairway, and the Gadeok fairway were 7.9, 19.1, 13.1, and 30.0 ships/h, respectively.

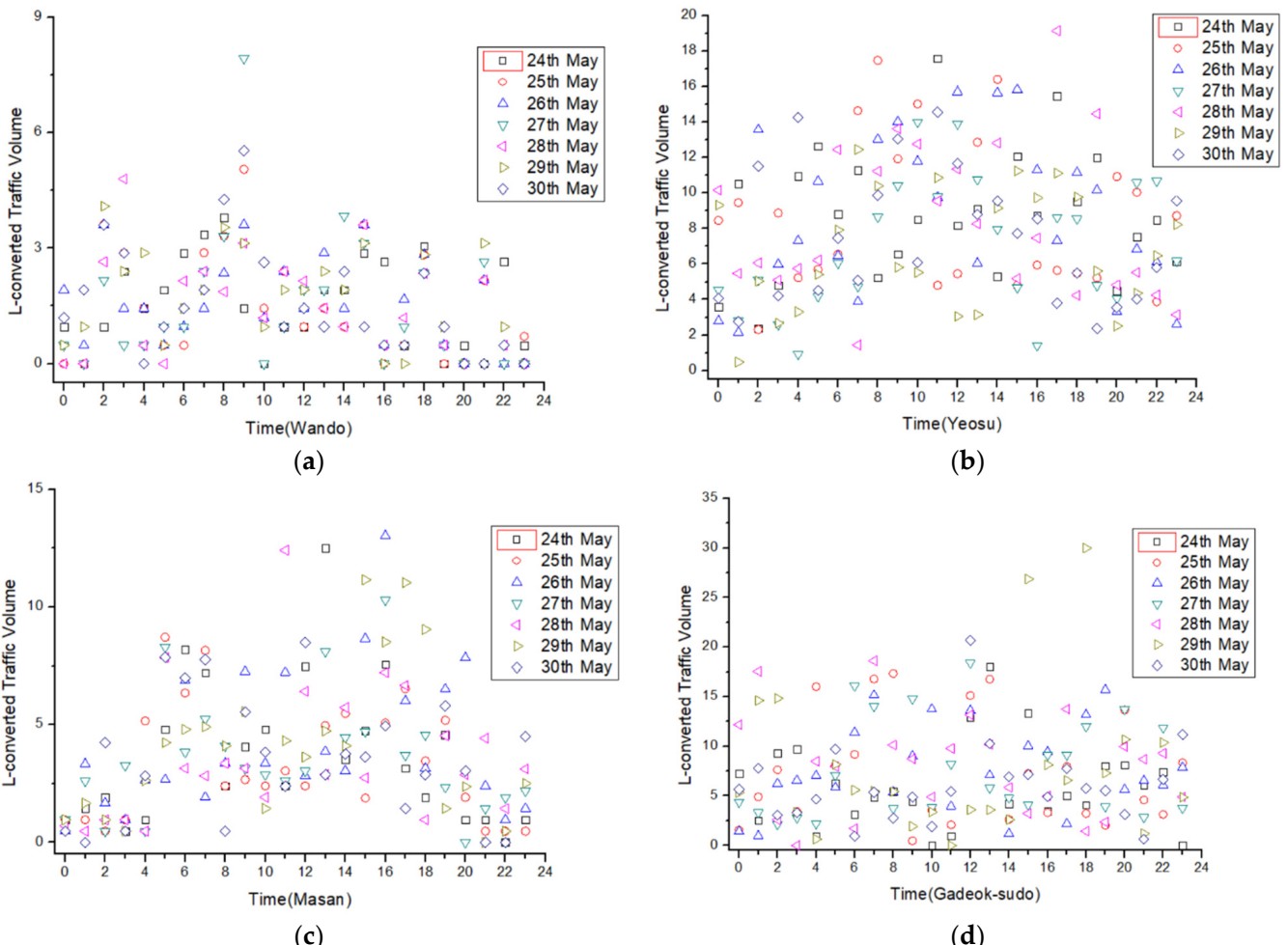

**Figure 5.** (**a**) L-converted traffic volume by time in the Wando fairway, (**b**) Yeosu area in Specific Traffic Safety Waters, (**c**) the Masan fairway, and (**d**) the Gadeok fairway.

*3.3. Analysis of Marine Traffic Congestion for Fairways*

In the Maritime Safety Act, the impact of maritime development projects on maritime traffic safety is investigated and evaluated in advance by the MTSA to eliminate risk factors. In addition, a maritime traffic congestion evaluation must be performed when designating or changing a fairway. Most safety diagnoses currently being conducted evaluate the degree of congestion using the bumper model [33].

The maritime traffic congestion assessment involves evaluating the current capacity by comparing the maximum traffic capacity that the fairway can accommodate with the traffic capacity actually being used. In MTSA, the number and size of ships passing the target fairway are investigated now and, in the future, and the results are presented after comparing the actual traffic volume with the maximum practical traffic volume that can be accommodated in the fairway based on this. Equation (3) shows the method to calculate the maritime traffic congestion [34,35].

$$T_C(\%) = \frac{Q_T}{Q_P} \times 100(\%) \tag{3}$$

where

| | |
|---|---|
| $T_C$ | Traffic Congestion; |
| $Q_T$ | Traffic Volume; |
| $Q_P$ | Practical Traffic Volume; |

Traffic capacity is divided into basic traffic capacity and practical traffic capacity. The basic traffic capacity means the maximum number of ships of a certain size per hour that can be accommodated when passing a waterway with a certain width at a constant speed. The basic traffic capacity refers to the maximum number of ships that reflects the area occupied by ships passing through the fairway as it is; thus, it indicates the maximum number of ships that can travel without a safety margin. However, because vessel navigation can be affected by factors such as sea and weather conditions, vessel navigation freedom, and maritime traffic management method, the actual applicable capacity is referred to as the practical transport capacity, and 20–25% of the basic transport capacity is applied [36].

The ship operator strives to maintain a certain safe distance from other ships, obstacles, and low-depth areas, to ensure the safety of their ship. This safe area around one's ship is called the occupied area of the vessel, and to evaluate the degree of traffic congestion, the occupied area of each vessel should be considered. When evaluating the vessel congestion in Maritime Traffic Safety Assessment, the occupied area of a ship is usually 8 L in the front and rear and 3.2 L on the side of the standard ship length in wide waters, and 6 L in the front and rear and 1.6 L on the sides in narrow waters [34]. In this study, an 82 m long vessel applied to the L-converted traffic volume was set as the standard vessel.

The basic traffic capacity can be obtained by dividing the product of the fairway width by the average ship speed on the fairway and size of the occupied area of the ship, as expressed in Equation (4) [35]:

$$Q = \frac{1}{\gamma s} WV \qquad (4)$$

where

| | |
|---|---|
| $Q$ | Basic Traffic Volume; |
| $\gamma$ | Long Diameter of occupied area of standard vessel (km); |
| $S$ | Short Diameter of occupied area of standard vessel (km); |
| $W$ | Fairway Width (km); |
| $V$ | Vessel Speed (km/h) |

Although each target fairway is near a port, because it is a relatively free navigation area, the ship's occupied area reflects the standard ship length of 8 L in front and rear and 3.2 L on the side. The traffic volume was thus analyzed. Table 5 shows the basic traffic capacity and the practical capacity for each target fairway.

**Table 5.** Basic traffic capacity and practical traffic capacity by destination fairway.

| Fairway | Speed (Knots) | Basic Traffic Volume (Number of Vessel/h) | Practical Traffic Volume (Number of Vessel/h) |
|---|---|---|---|
| Wando | 10 | 245.1 | 61.3 |
| | 12 | 294.1 | 73.5 |
| | 15 | 367.6 | 91.9 |
| Yeosu | 10 | 295.3 | 73.8 |
| | 12 | 354.3 | 88.6 |
| | 15 | 442.9 | 110.7 |
| Masan | 10 | 245.1 | 61.3 |
| | 12 | 294.1 | 73.5 |
| | 15 | 367.6 | 91.9 |
| Gadeok | 10 | 248.0 | 62.0 |
| | 12 | 297.6 | 74.4 |
| | 15 | 372.1 | 93.0 |

Since the speed of a ship differs based on the ship size, it is difficult to apply a specific speed. To compare whether the degree of congestion varies for each ship speed, we analyzed the congestion levels for 10 kts, 12 kts, and 15 kts. Based on the results of the maritime traffic survey conducted in Section 3, the congestion level according to the traffic

capacity for each fairway in Table 5 was analyzed, and the congestion level per hour for 7 days is shown in Figure 6.

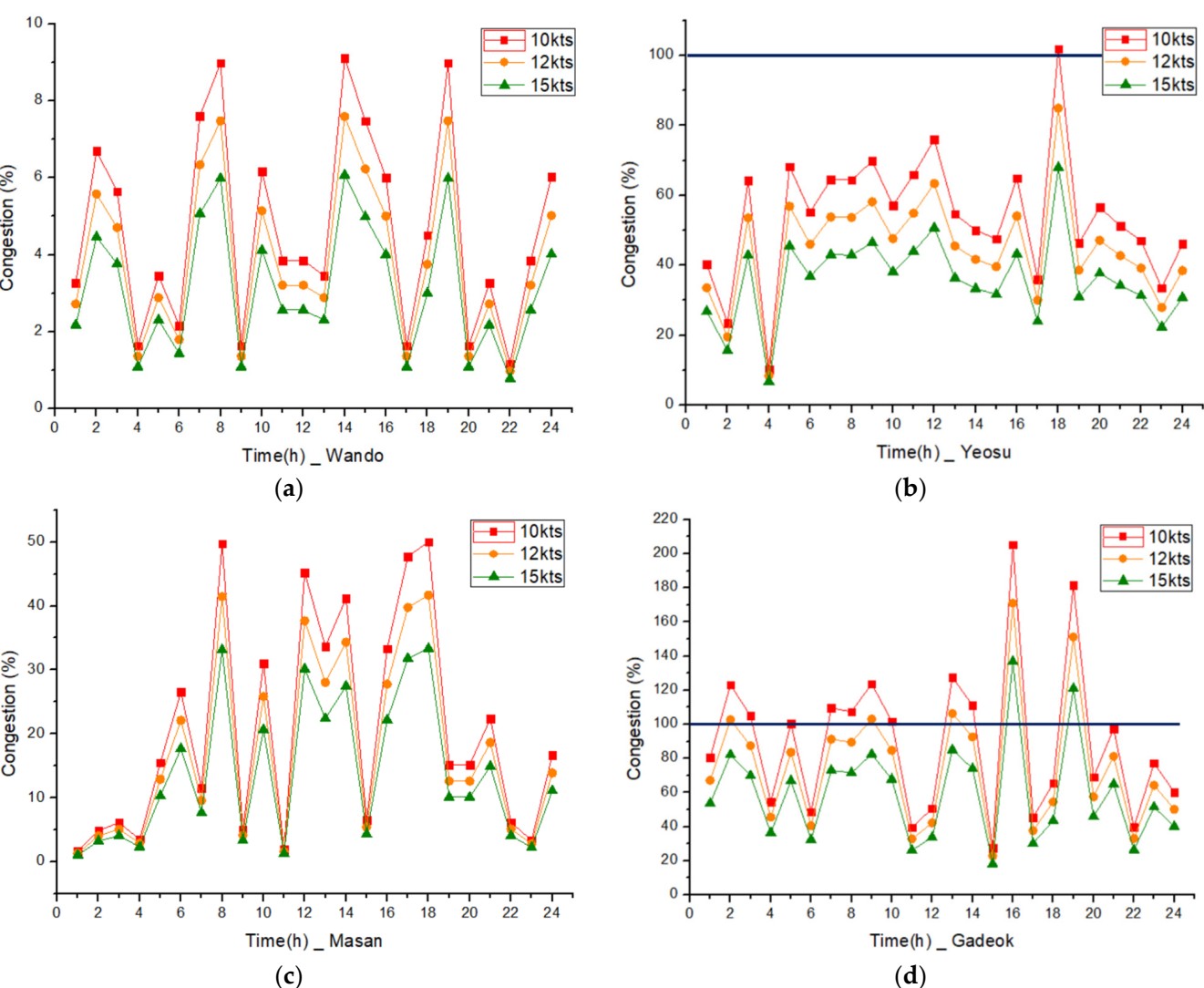

**Figure 6.** (**a**) Hourly congestion evaluation result of the Wando fairway, (**b**) the Yeosu fairway, (**c**) the Masan fairway, and (**d**) the Gadeok fairway.

As the degree of congestion is a comparison of the actual traffic capacity with the actual maritime traffic volume, 100% means that ships are passing at the practical traffic capacity, and because the fairway is saturated, the traffic of ships is concentrated at a specific time. Thus, it is necessary to disperse the constant mass. According to Figure 3b,d, we found that congestion on the Yeosu fairway and Gadeok fairway was over 100% at a specific time; thus, it was analyzed that it is necessary to solve the congestion of the fairway. Table 6 shows the average hourly congestion and peak-time congestion evaluation results for each fairway.

On the Wando fairway, the average hourly congestion level was up to 1.4%, and the congestion level during the hour with the most traffic was 9.1%, indicating considerable leeway in the fairway capacity. In the Yeosu fairway, the average hourly congestion level was up to 25.5%, and the congestion level during the busiest time period was 101.9%. The average congestion of the Masan fairway was up to 6.5% per hour, and the congestion level during the busiest hour was 50.0%, indicating a certain amount of leeway. In the Gadeok fairway, the average hourly congestion level was the highest, at 36.9%, the highest among the target fairways, and the congestion level during the busiest time period was

205.2%, which was analyzed to significantly exceed the practical traffic capacity. However, because there is still plenty of traffic in other time zones, it may be a short-term measure to reduce congestion by distributing traffic to other time zones. However, this cannot be regarded as a fundamental solution; furthermore, if the traffic volume increases and ships become larger due to port development in the future, the probability of marine accidents may increase considerably, depending on the congestion level of the fairway.

**Table 6.** Average Hourly Congestion and Peak-Time Congestion.

| Fairway | $L^2$-Converted Traffic Volume (Number of Vessel/h) | Speed (Knots) | Traffic Congestion (Average) | Traffic Congestion (Peak-Time) |
|---------|---------|---------|---------|---------|
| Wando | 0.86 | 10 | 1.4% | 9.1% |
|  |  | 12 | 1.2% | 7.6% |
|  |  | 15 | 0.9% | 6.1% |
| Yeosu | 18.8 | 10 | 25.5% | 101.9% |
|  |  | 12 | 21.2% | 84.9% |
|  |  | 15 | 17.0% | 67.9% |
| Masan | 4.0 | 10 | 6.5% | 50.0% |
|  |  | 12 | 5.4% | 41.7% |
|  |  | 15 | 4.4% | 33.4% |
| Gadeok | 22.9 | 10 | 36.9% | 205.2% |
|  |  | 12 | 30.7% | 171.0% |
|  |  | 15 | 24.6% | 136.8% |

*3.4. Conformity Review*

In Section 2, we analyzed the current status and design criteria of the target fairway. The widths of the Wando fairway, the fairway within the Yeosu area, the Masan fairway, and the Gadeok fairway were 1660, 2000, 1660, and 1680 m, respectively. According to the PIANC Guide, the width of the fairway requires 3.8~18.6 times the width of the largest vessel, and at least 1~2 times the length of the largest vessel, according to the port and fishing port design standards. As a result of the maritime traffic survey analyzed in this chapter, the largest vessel on the Wando fairway is 161 m long and 25 m wide, that on the Yeosu fairway is 400 m long and 59 m wide, that on the Masan fairway is 345 m long and 53 m wide, and that on the Gadeok fairway is 400 m long and 59 m wide.

Therefore, we found that the required fairway width according to the PIANC Guide was 95~465 m based on the passing ships of the Wando fairway and 161~322 m according to the port and fishing port design standards, and the current fairway width satisfies the design standards. The required fairway width according to the PIANC Guide was 224~1097 m based on the vessels passing through the Yeosu fairway and 400~800 m according to the port and fishing port design standards. Furthermore, the required fairway width according to the PIANC Guide was 201~986 m based on the passing ships of the Masan fairway, and that according to the harbor and fishing port design standards was 345~690 m, which satisfies the design standards. The required fairway width according to the PIANC Guide was 224~1097 m based on the passing ships of the Gadeok fairway and 400~800 m, according to the port and fishing port design standards.

According to the fairway design standards based on the largest vessel, such as the PIANC Guide or the port and fishing port design standards, the current fairway width is sufficient. In addition, excess fairways were found. The average hourly congestion of the Yeosu fairway was up to 25.5%, and the congestion level during the busiest time period was 101.9%, indicating that there was room the entire time. In addition, the average hourly congestion level of the Gadeok fairway was up to 36.9% and the congestion level during the busiest time period was 205.2%, significantly exceeding the traffic capacity of the fairway. Distributing traffic from the time when ships are concentrated to other time zones may be a

short-term measure; however, in the long run, fundamental solutions, such as widening the fairway, are required.

As shown in the analysis of previous studies, fairways are installed in a sea area where collisions and grounding accidents are highly probable due to a high density of vessel traffic or low-depth areas, as in Resolution A. 572(14) adopted by the IMO [16–18]. Therefore, when evaluating the design of a new fairway or the adequacy of an existing fairway, it is necessary to consider the risk of collision and grounding.

Section 4 examines the adequacy of the target fairway using the ship collision risk model and the grounding risk model, and proposes a method to design the appropriate fairway width, considering the amount of ship traffic.

## 4. Re-Designing of the Appropriate Fairway Width

In this chapter, based on the risk assessment model—the ES model—the collision risk for the traffic flow of a vessel is evaluated, and the traffic distribution and grounding frequency of the seabed topography are evaluated through the IWRAP Mk II model, to design an appropriate fairway width. In Sections 2 and 3, we reviewed the design criteria of PIANC and port and fishing port design criteria, which are mainly evaluated when MTSA designates or changes fairways, and the maritime traffic congestion was evaluated. Design standards based on the maximum sailing vessel, such as in the PIANC Guide or the port and fishing port design standards, evaluate the current fairway width as appropriate, but in the congestion assessment based on the traffic volume, some fairways were analyzed as exceeding the traffic capacity of the fairway.

In this chapter, we propose a method to identify the appropriate fairway width, by evaluating each target fairway based on the ship's collision probability and grounding probability, and reducing or increasing the fairway width in directions where room is available.

### 4.1. Assessment Overview

Various models are used to evaluate the risk of collision and grounding. In this study, the ES model was used for the collision risk and the IALA MkII model was used for the grounding risk model.

The risk model related to maritime traffic and operation can be separated into a traffic status evaluation model, a navigation difficulty evaluation model, and an IALA evaluation model. The evaluation model can be applied according to the variables of each element, and, as in this study, it is appropriate to apply the navigation difficulty evaluation model to analyze the psychological effect of the fairway width on the operator. Operational difficulty evaluation models include the SJ model, BC model, ES model, PARK model, and other models [37]. Among them, the ES model is a model that quantitatively expresses the traffic situation of nearby ships and the degree of risk for navigation obstacles. In the Maritime Traffic Safety Assessment, changes in the maritime traffic environment due to new fairways and facility construction are evaluated using the ES model. Therefore, in this study, collision risk was evaluated using the ES model.

The evaluation related vessel grounding was first proposed in the 1980s, and different models are applied depending on the research purpose and variables. Arsham Mazaheri and Jutta Ylitalo (2010) divided the grounding risk assessment model into two groups: an analytical model and a statistical model. The analytical model includes Fujii's model and Macduff's model, and the statistical model includes the Pedersen Mmodel and Simonsen Model. Among them, the Pedersen and Simonsen models have been the most used recently and are the basis of the current IWRAP MkII of IALA. Although it is difficult to say that a specific model provides more accurate results than other models, in previous studies, Fujii's model and Macduff's Mmodel were suggested to have weaknesses in probability analysis, and the Simonsen model was evaluated to be more reasonable than the Pedersen Model [38]. Therefore, in this study, IWRAP MkII, to which the statistical model was applied, was used as a grounding risk analysis model.

In this chapter, the concept and theoretical background of the ES model, used to evaluate the collision risk of a ship, and the concept and theoretical background of the IWRAP Mk II model, to evaluate the frequency and probability of a ship's grounding, are reviewed. Furthermore, to ensure the reliability of the evaluation, an evaluation scenario was established and presented.

1. ES Model

The environmental stress (ES) model was developed to classify the marine traffic environment surrounding a ship into an operating environment and the sea area environment, and to quantitatively evaluate the degree of stress applied to the ship operator by these two environments. The ES model considers the bearing and distance to obstacles and the speed between ships in the range of $\pm 90°$ centering on the course of the ship, calculates the time margin, and quantifies the time margin as a stress value felt by the ship operator. The amount of environmental stress that a ship operator experiences is called the ES value. The environmental stress value consists of an ES value for land ($ES_L$), which is the amount of stress caused by the marine environment, such as topography or facilities, and the ES value for ships ($ES_S$), which is the amount of stress caused by the operating environment, such as other ships. The stress values are aggregated and called the aggregation of ES ($ES_A$) values. The ES stress is classified into four levels (0–500, 500–750, 750–900, and 900–1000). If the stress level is 750 or more, it is evaluated as an acceptable limit, and if it is 900 or more, it is evaluated as unacceptable [37,39,40].

In this study, the $ES_A$ value was used to evaluate the risk of collision between ships passing a fairway and the risk due to the operation of the ship within a limited fairway.

First, the $ES_L$ calculation method is given in Equations (5) and (6) [37].

$$ES_L = \sum_{\varphi=-90°}^{+90°} SJ_L \tag{5}$$

$$SJ_L = \alpha \times (R/V) + \beta \tag{6}$$

where

$SJ_L$      Risk of obstacles;
$R$      Distance to obstacles;
$V$      Speed of the own ship;
$\alpha, \beta$      Coefficient determined by natural conditions;
$\alpha = -0.00092 \times \log 10(GT) + 0.0099$
$\beta = -3.82$.

The calculation method of $ES_S$ is as Equations (7) and (8) [37].

$$ES_S = \sum_{\varphi=-90°}^{+90°} SJ_S \tag{7}$$

$$SJ_S = \alpha \times (R/V \times V/L_m) + \beta = \alpha \times (R/L_m) + \beta \tag{8}$$

where

$\alpha = 0.00192 \times L_m$
In case the condition of encounter with the target ship is:
Crossing from starboard, $\beta = -0.65 \times \ln Lm - 2.07$.
If crossed from portside, $\beta = -0.65 \times \ln Lm - 2.35$.
If meeting from the bow, $\beta = -0.65 \times \ln Lm - 2.07$.
If the ship is overpassed from the stern, $\beta = -0.65 \times \ln Lm - 0.85$.
$SJ_S$      Risk concerning the target ship;
$R$      Relative distance between own ship and target ships;
$V$      Relative speed between own ship and target ships;
$Lm$      Average length of the ship and target ships.

Equation (9) is a formula for the comprehensive evaluation of $ES_L$ and $ES_S$ [37].

$$ES_A = \sum_{\varphi=-90°}^{+90°} \max\{SJ_L, SJ_S\} \tag{9}$$

2.  IWRAP Grounding Probability

IALA Waterway Risk Assessment Program (IWRAP Mk II)—a maritime risk assessment tool—quantifies, predicts, and evaluates risks related to maritime traffic. IWRAP Mk II is used as a useful modeling tool for maritime risk assessment and estimates the frequency of collisions and grounding with ships, based on traffic volume, traffic distribution, and seabed topography information [41].

In IWRAP, the types of situations in which grounding may occur are the following: ① A vessel navigating at a normal speed on a normal fairway caused by human error, ② a vessel that fails to change course at a changing point near an obstacle, ③ a vessel that takes an evasive action near an obstacle. The ships on grounding accident are classified into types ①, ②, and ③, including the types of accidents, ④ loss of propulsion, etc., and the formula applied according to the type of situation is also different.

In this study, only categories 1 and 2 are reflected in the evaluation to obtain the probability of grounding in the ship's operation and changing direction under normal circumstances [42,43]. The method of calculating the grounding probability for categories 1 and 2 was as shown in Equations (10) and (11):

$$N_I = \sum_{Ship\ Class,\ i} P_{c,i} Q_i \int_{z_{min}}^{z_{max}} f_t(z) dz \tag{10}$$

$$N_{II} = \sum_{Ship\ Class,\ i} P_{c,i} Q_i \exp(-d/a_t) \int_{z_{min}}^{z_{max}} f_t(z) dz \tag{11}$$

where

| | |
|---|---|
| $a_t$ | Average distance between position checks by the navigator; |
| $d$ | Distance from the obstacle to the bend in the navigation route varying with the lateral position, s, of the ship; |
| $i$ | Index for ship class, categorized after vessel type and dead weight or length; |
| $f_t(z)$ | Probability density function for the ship traffic; |
| $N_I$ | Expected number of category I grounding events per year; |
| $N_{II}$ | Expected number of category II grounding events per year; |
| $P_{c,i}$ | Causation probability, i.e., ratio between ships grounding and ships on a grounding course; |
| $Q_i$ | Number of ships in class i passing a cross section of the route per year; |
| $z$ | Coordinate in the direction perpendicular to the route; |
| $z_{min}, z_{max}$ | Transverse coordinates for an obstacle. |

In this study, based on the GICOMS data obtained for 7 days, the frequency of grounding accidents with obstacles near each target fairway was estimated reflecting the maritime traffic flow.

3.  Assessment Standard and Scenario

To set the appropriate fairway width by analyzing the risk of collision and grounding of the existing fairways, the actual fairway width and distance from obstacles in Figure 3 were set. For ships passing the fairway, the collision risk was evaluated through the ES risk model, and the grounding probability was evaluated using the IWRAP grounding frequency calculation model for navigation obstacles, such as islands or low water depths outside the fairway. The ship traffic volume, speed, and size were composed based on traffic survey results, and ship traffic volume is reflected by setting the peak time period of the day with the most traffic. For ease of evaluation, the concept of L-converted traffic was applied to vessel traffic.

Figure 7 shows the scenarios set up to analyze the collision risk and the grounding probability for each vessel passing the fairway, and Table 7 shows the parameter details for the scenarios.

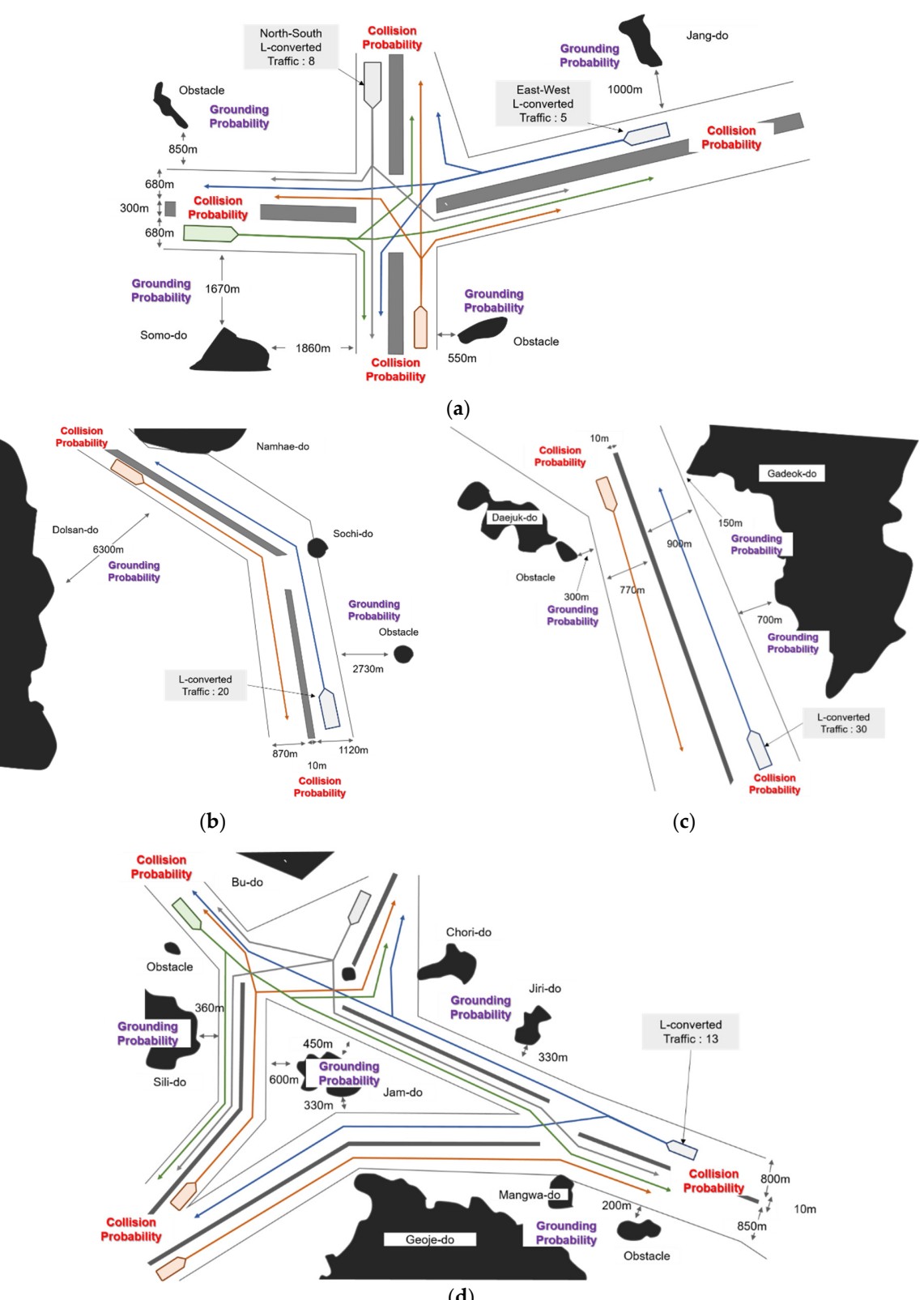

**Figure 7.** (**a**) Collision risk and grounding probability evaluation scenario of the Wando fairway, (**b**) the Yeosu fairway, (**c**) the Gadeok fairway, and (**d**) the Masan fairway.

**Table 7.** Details of collision and grounding risk assessment parameters.

| Type | Wando | Yeosu | Masan | Gadeok |
|---|---|---|---|---|
| Traffic Volume | 8/h | 20/h | 13/h | 30/h |
| Ship's Speed | | 10 knots, 12 knots, 15 knots | | |
| Width of Fairway (minimum) | 680 m | 870 m | 800 m | 770 m |
| Distance from Obstacles (minimum) | 550 m | Adjacent to the fairway | 200 m | 150 m |

Figure 7a shows the collision and grounding risk assessment scenario for the Wando fairway. Scenarios were set up to allow ships to navigate in all directions according to the characteristics of the fairway, where vessels travelling in the east–west and north–south directions crossed. In addition, obstacles such as Jangdo on the upper side of the ship passing east–west, and obstacles at a distance of 550 m below the north–south fairway, were reflected in the grounding probability evaluation scenario. According to the results of the maritime traffic survey, 8 L-converted traffic per hour was set to pass. Figure 7b shows the evaluation scenario for the Yeosu fairway. There is a risk of collision and grounding due to the presence of Sochido near the right side of the fairway, and the scenario reflected this. According to the results of the maritime traffic survey, 20 L-converted traffic per hour was set to pass. Figure 7c shows the evaluation scenario for the Gadeok fairway. The width of the left passage is 770 m, and that of the right passage is 900 m. The obstacle closest to the right passage is 150 m to the west end of the Gadeok Island, and the obstacle closest to the left passage is 300 m to the right end of Daejuk Island. According to the results of the maritime traffic survey, we found an L-converted traffic volume of 30 ships per hour. Figure 7d shows an evaluation scenario for the Masan fairway. As shown in the previous analysis, the Masan Passage is a complex water area with vessels passing through Jinhae Port, Masan Port, and Tongyeong Port. According to the results of the maritime traffic survey, it was established that there is an L-converted traffic volume of 13 ships per hour.

In the ES model applied to evaluate the collision risk of a ship, the risk can be evaluated according to the past, present, and future traffic flows by setting various fairways and generating variables. In the maritime traffic safety diagnosis according to the Maritime Safety Act, this evaluation is used to designate new fairways and change existing fairways. In the ES model, the overall risk of the sea area is evaluated as the ratio of the section ES value over 750, where the ES value exists, to the entire section. When the ratio of the ES value of 750 or higher is 10% or more of the total, it is judged that safety measures are necessary [29,44]. Therefore, in this study, when the ratio of ES Value of 750 or higher was 10% or more of the total, it was determined that a risk of collision exists, and it was set as a comparative index. Meanwhile, the IWRAP Mk II model was applied to evaluate the risk of grounding with a navigation obstacle. This model presents the results of the frequency of groundings that occur during one year in the maritime traffic environment and vessel navigation conditions. However, the frequency of grounding can vary from high to low risk, depending on the amount of vessel traffic; furthermore, it is not possible to know whether the sea area has a high risk of grounding when the frequency is quantitatively high. In the maritime traffic safety diagnosis of the Maritime Safety Act, to evaluate the risk of collision with a specific point during a ship simulation evaluation, the existence of collision risk is judged based on the nearest voyage distance. Assuming that the navigational range of a general ship follows a normal distribution, the collision probability can be calculated using the deviation of each nearest navigation distance. In addition, the AASHTO Guide (2009) provides a method for determining the design vessel for the design of offshore bridge structures. The design ship is calculated so that it is within the allowable standards, according to the importance of the bridge [45,46]. In the standard, the allowable standard for AF is 0.0001 for important bridges and 0.001 or less for general bridges. When these criteria were applied in this study mutatis mutandis and when the total ship traffic volume

is aggregated based on the frequency of grounding, if the frequency of grounding is higher than the probability of 1 in 10,000, a risk of grounding is considered to exist.

### 4.2. Results of ES Model Simulation

In this chapter, the sea traffic flow according to the scenario in Figure 7 is reproduced using the ES model, based on the sea traffic volume analyzed in Section 3. Based on the amount of traffic passing through each fairway, the speed was changed to 10 kts, 12 kts, and 15 kts, and evaluation was performed using the ES Model. Figure 8 shows the visualization of the evaluation results of the ES model for each fairway.

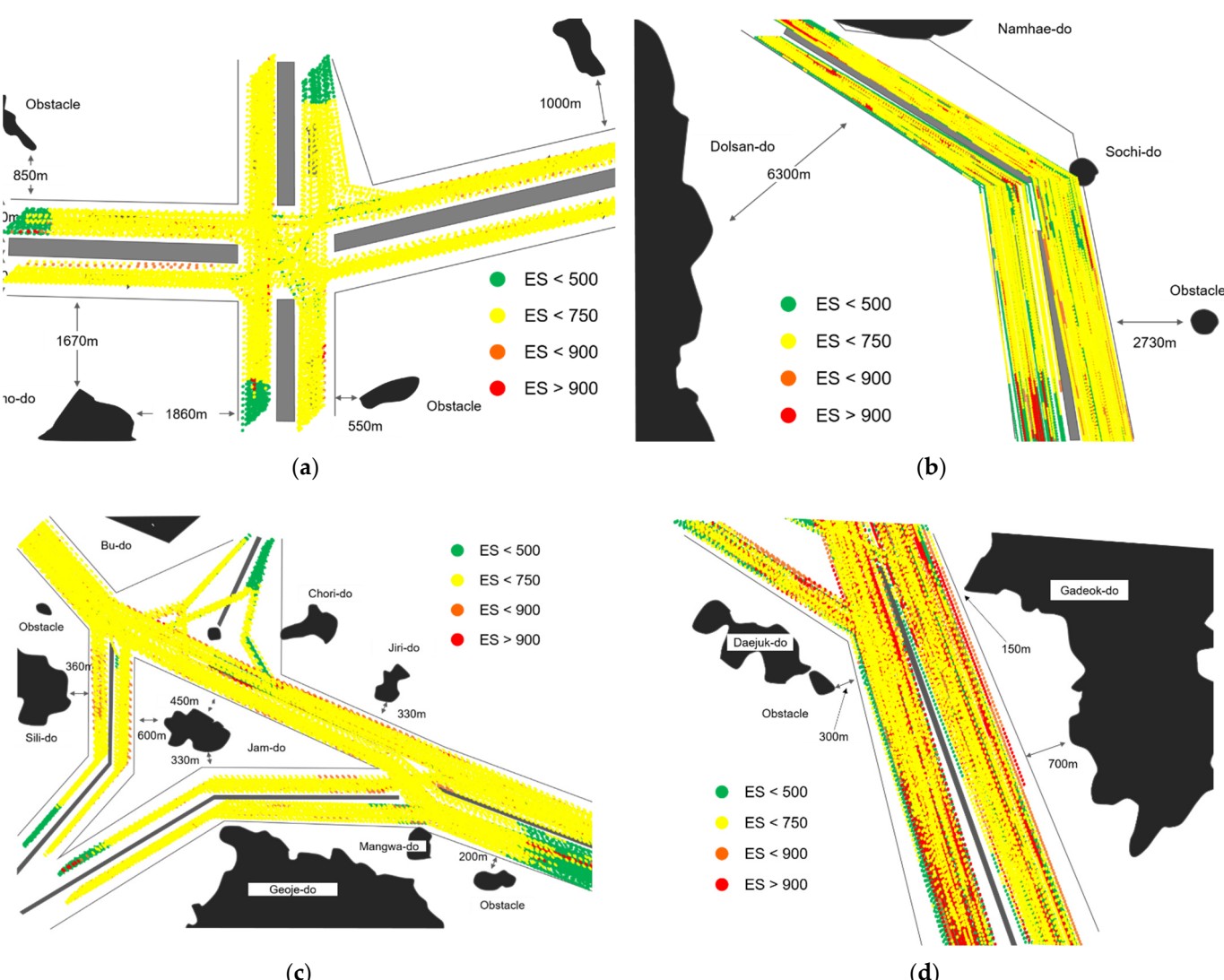

**Figure 8.** (**a**) Collision risk evaluation result for the Wando fairway, (**b**) the Yeosu fairway, (**c**) the Masan fairway, and (**d**) the Gadeok fairway.

Green indicates ES values less than 500, yellow indicates ES values less than 750, orange indicates ES values less than 900, and red indicates ES values of 900 or more.

Table 8 shows the values of each $ES_S$, $ES_L$, and $ES_A$, which is the overall environmental stress, when the ES value for each fairway is the highest.

**Table 8.** ES value for each destination fairway.

| ES Value | Wando | | | Yeosu | | | Masan | | | Gadeok | | |
|---|---|---|---|---|---|---|---|---|---|---|---|---|
| | $ES_A$ | $ES_L$ | $ES_S$ | $ES_A$ | $ES_L$ | $ES_S$ | $ES_A$ | $ES_L$ | $ES_S$ | $ES_A$ | $ES_L$ | $ES_S$ |
| 0~500 | 14.1 | 16.1 | 97.4 | 49.9 | 69.5 | 86.2 | 13.6 | 15.4 | 94.1 | 26.0 | 64.9 | 57.8 |
| 500~750 | 79.8 | 83.7 | 1.1 | 33.6 | 28.9 | 5.8 | 72.6 | 80.0 | 1.8 | 30.9 | 34.4 | 12.5 |
| 750~900 | 4.4 | 0.2 | 0.2 | 7.6 | 1.6 | 1.4 | 9.2 | 4.5 | 0.6 | 12.2 | 0.6 | 4.1 |
| 900~1000 | 1.7 | 0.0 | 1.3 | 8.9 | 0.0 | 6.6 | 4.7 | 0.0 | 3.5 | 30.9 | 0.0 | 25.6 |

In the case of the Wando fairway, 1.5% of the total ships had an ES value of 750 or higher and 0.2% had an ES value of 750 or higher for the marine environment. The overall ES level was 6.1% of the total. The risk was analyzed to be low. In the case of the Yeosu fairway, 8.0% of the cases had an ES value of 750 or higher for the ship operating environment and 1.6% of the cases had an ES value of 750 or higher for the marine environment. The burden on the operator is high when operating the target fairway. In the case of the Masan fairway, the ES value for the ship operation environment of 750 or higher was 4.1% of the total, and the ES value for the sea area environment at 750 or higher was 4.5% of the total. It was analyzed that the operational burden of the operator on this fairway was slightly high, and the overall environmental stress level was 13.9% of the total, which was lower than that of the Yeosu fairway. In the case of the Gadeok fairway, 29.7% of cases with an ES value of 750 or higher for the vessel operating environment and 0.6% of the total cases with an ES value of 750 or higher for the marine environment were found. Since it has many fairways, it was analyzed that the risk of passage with other ships was very high, and the overall ES level was 43.1% of the total.

Figure 9 is a separate visualization of only the parts where the ES value is 750 or higher, to analyze which section of each fairway has the highest burden on the operator.

Overall, the risk of the Wando fairway was analyzed to be low. In the case of the Yeosu fairway, the ES value was high in the north of Sochido, especially in the right fairway. In the case of the Masan Passage, the risk was high in the place where vessels entered and crossed the Masan Passage on the east side and where they passed along the north fairway. In particular, the risk was somewhat high in the north and west of Jangdo. In the case of the Gadeok fairway, the overall risk was found to be very high.

*4.3. Results of Grounding Probabilities*

Based on the maritime traffic flow and traffic volume analyzed in Section 3, the frequency of grounding of IWRAP Mk II was derived by applying the maritime traffic environment according to the scenario shown in Figure 7. By reflecting the characteristics of each fairway, obstacles and low-depth sections in the vicinity were also reflected. Figure 10 highlights the results of the estimated frequency of grounding for each fairway.

The closest island, reef, and low-water zone were set as the grounding standard. In the case of the Yeosu fairway, because there is a marina and a low-depth section on the right side of the fairway, the fairway was set as the grounding standard on the right side, and the grounding standard was set on the left side of the fairway, according to the water depth for each draft. Table 9 shows the grounding probabilities based on the estimated grounding frequency and traffic volume.

**Table 9.** Probability of grounding by fairway.

| Name of Fairway | Traffic Volume (Actual/7 Days) | Traffic Volume (Estimated/Year) | Predicting Grounding Frequencies | Grounding Probability |
|---|---|---|---|---|
| Wando | 779 | 40,619 | 0.0294852 | $7.2590 \times 10^{-7}$ |
| Yeosu | 1274 | 66,430 | 0.146378 | $2.2035 \times 10^{-6}$ |
| Masan | 957 | 49,901 | 0.302582 | $6.0636 \times 10^{-6}$ |
| Gadeok | 1002 | 52,247 | 0.107486 | $2.0573 \times 10^{-6}$ |

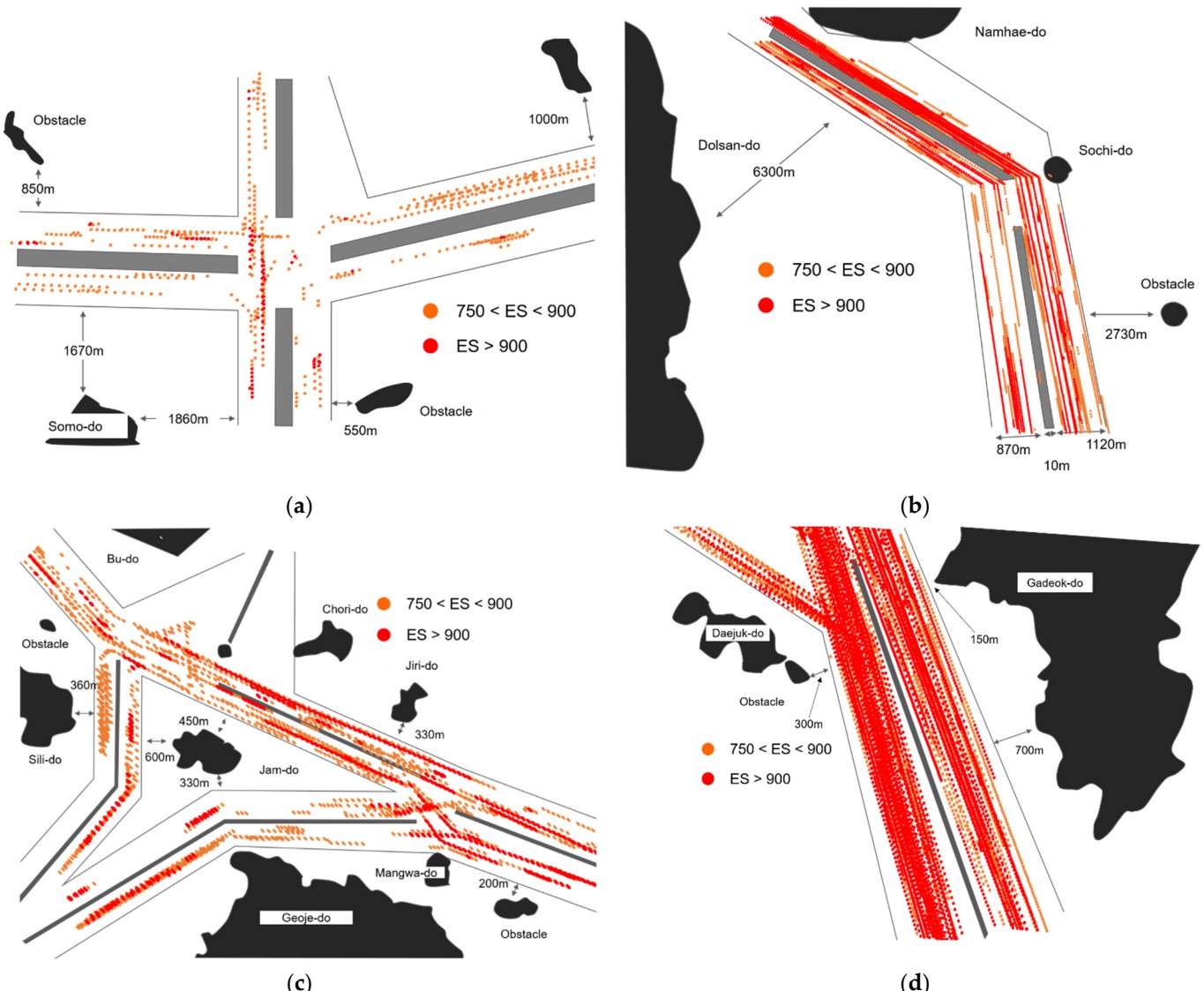

**Figure 9.** (**a**) Distribution map of ES ≥ 750 for the Wando fairway, (**b**) the Yeosu fairway, (**c**) the Masan fairway, and (**d**) the Gadeok fairway.

Following the evaluation criteria established in Section 4.1, we judged that there exists a risk of grounding when the probability of grounding exceeds $10^{-4}$ when the frequency of grounding for the total vessel traffic is prepared. Based on the 7-day traffic data, when the estimated frequency of grounding for each fairway was calculated from the estimated ship traffic for one year, the Wando fairway showed a grounding probability of $7.2590 \times 10^{-7}$, the Yeosu fairway had a grounding probability of $2.2035 \times 10^{-6}$, the Masan fairway had a grounding probability of $6.0636 \times 10^{-6}$, and the Gadeok fairway had a grounding probability of $2.0573 \times 10^{-6}$. The grounding probability was found to be lower than $10^{-4}$ for all fairways subject to evaluation; thus, the risk of grounding was found to be good in the current traffic and sea area conditions.

### 4.4. Results of ES Model Simulation for Fairway Expansion

As a result of analyzing the risk of ship collision according to the ES Model and the risk of grounding according to IWRAP Mk II for the target fairway, we found that the risk of collision was high in the Yeosu fairway, Masan fairway, and Gadeok fairway, whereas the grounding risk was good. In this study, the evaluation of the grounding probability was based on islands, obstacles, and low water areas farther than the end line of the fairway,

without considering the fairway line. Therefore, the fairway can be expanded to some extent to the islands, obstacles, and low-depth areas, which form the standard of grounding for each fairway. However, if the vessel's navigation area spreads widely as the fairway expands, the distribution is formed based on the center of the fairway, and assuming a normal distribution, the grounding probability may increase to a certain extent, depending on the deviation. Since the evaluation of the grounding probability in this study is based on actual data, the track distribution according to the fairway expansion, the ship navigation deviation, and the estimate of the grounding probability will be left for future research.

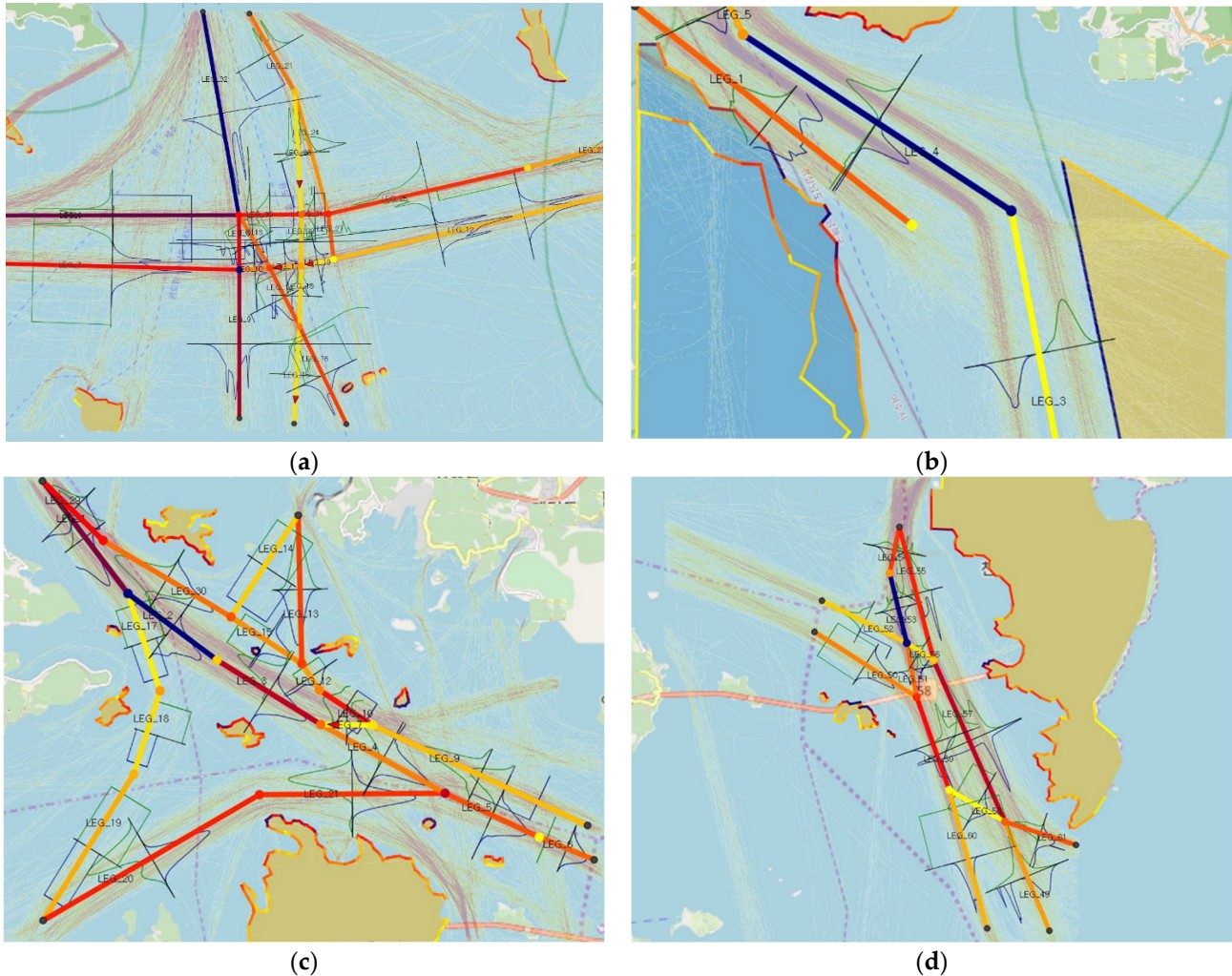

(a)  (b)

(c)  (d)

**Figure 10.** (**a**) Grounding frequency estimation results for the Wando fairway, (**b**) the Yeosu fairway, (**c**) the Masan fairway, and (**d**) the Gadeok fairway.

In this chapter, for fairways with high collision risk as observed using the ES model, such as the Yeosu fairway, Masan fairway, and Gadeok fairway, the fairway was extended to the safe water area between the fairway and obstacles, and the risk was re-evaluated to examine any changes. As the Yeosu Passage has an anchorage and a low-depth section on the right, and as there is a free water area on the left, the fairway was expanded and reset by approximately 1 km to the left. The Masan Passage was extended by approximately 300 m to Jiri Island, which is located above the North Passage, and based on Jamdo in the middle of the Passage, the North Passage was extended by approximately 400 m, the West Passage by approximately 500 m, and the South Passage by approximately 300 m. The Gadeokdo Island expanded the fairway by approximately 100 m in the direction of Gadeokdo on the right and approximately 300 m up to the obstacle on the left. Based on the ES Model in Ch. 4.2, the amount of risk change before and after fairway expansion was

analyzed. Figure 11 shows the current situation before the expansion of the fairway and the change in the ES value after the expansion.

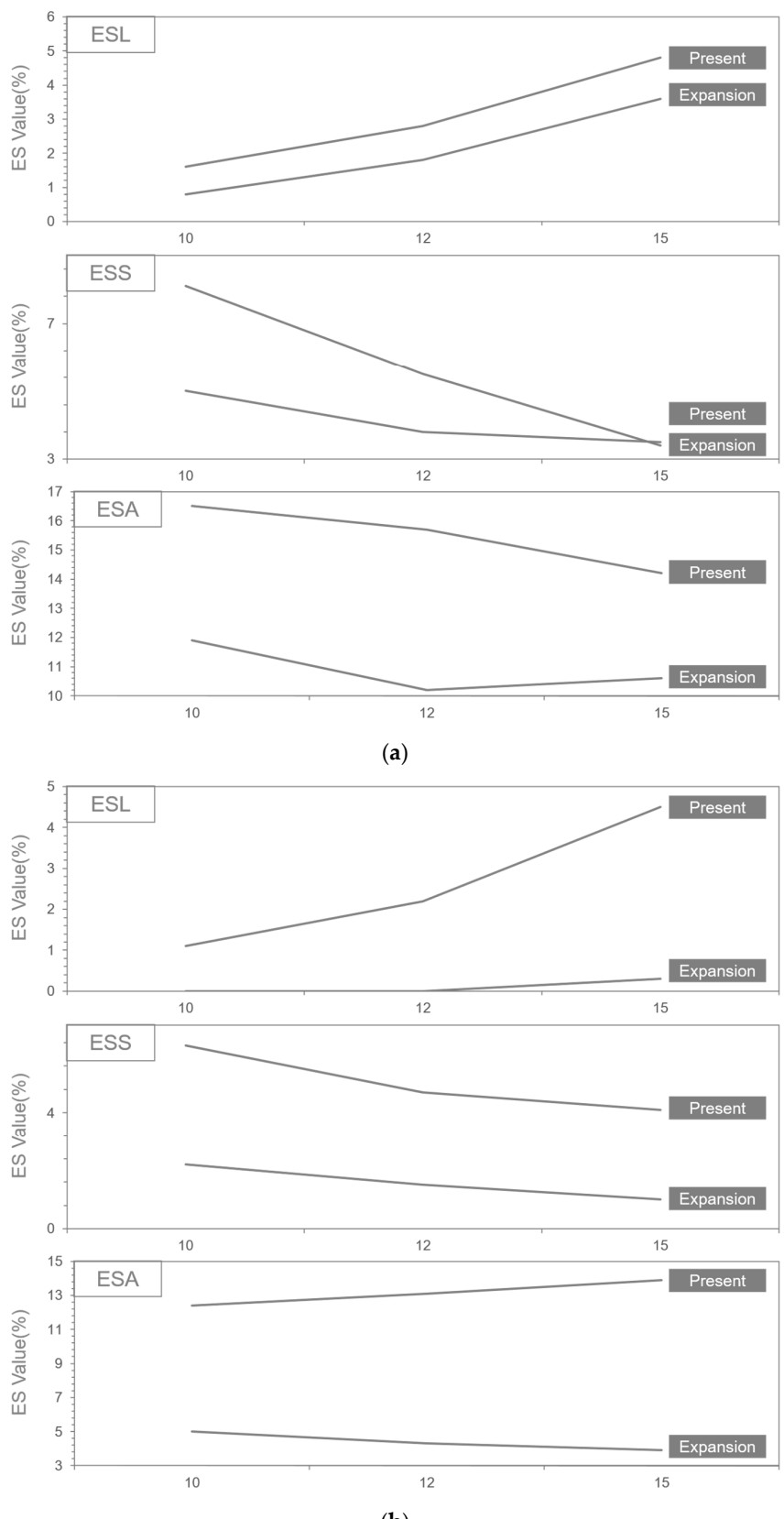

**Figure 11.** *Cont.*

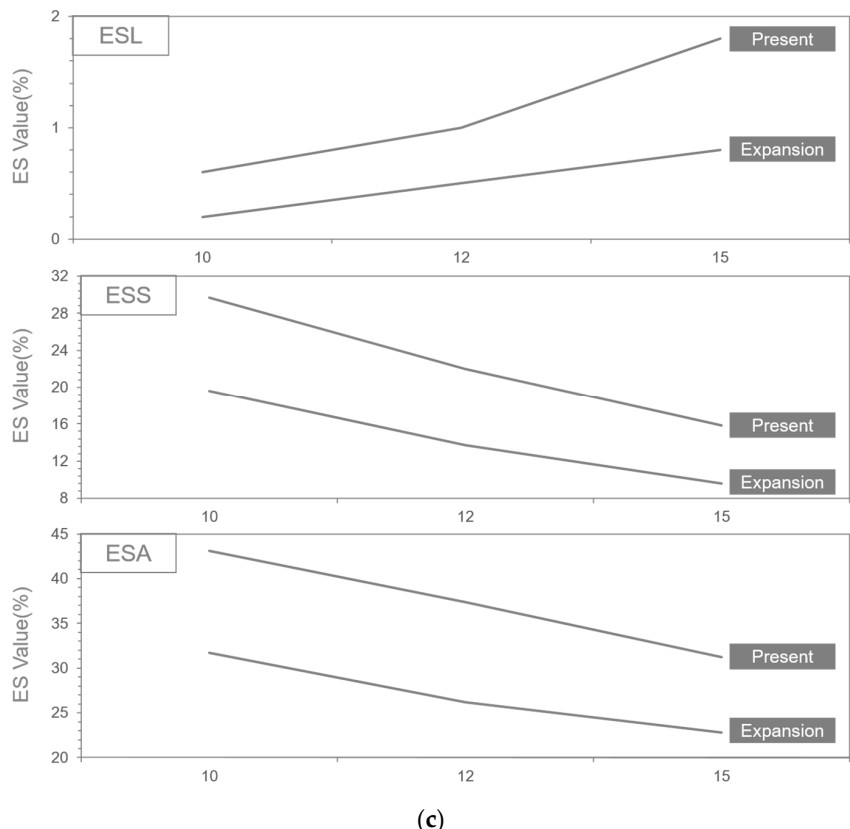

(**c**)

**Figure 11.** (**a**) Analysis of the ES value before and after expansion of the Yeosu fairway, (**b**) the Masan fairway, and (**c**) the Gadeok fairway.

In all three fairways, the ES Value decreased when the fairways were expanded overall. As for the characteristics of $ES_L$ and $ES_S$, $ES_L$ increased and $ES_S$ decreased as the vessel speed increased. If we consider whether $ES_A$, which is a comprehensive environmental stress, follows the trend of $ES_S$ or $ES_L$ as the ship speed increases, it can be seen whether the fairway has a large influence on the vessel traffic volume, or whether the characteristics of the fairway affect the ship safety. In the case of the Figure 11a Yeosu fairway and Figure 11c Gadeok fairway, the $ES_A$ value was found to be greatly affected by the vessel traffic volume, as the ES value decreased as the vessel speed increased. In particular, in the case of the Gadeok fairway, the ES value of $ES_A$ decreased overall; however, the shape of the $ES_A$ graph was similar to that of the $ES_S$, indicating that the amount of traffic of the vessel had a great influence on the results. In contrast, in the case of the Masan fairway in Figure 11b, the ES value of the $ES_A$ increased as the ship speed increased in the current state, and the fairway was found to be significant affected by the fairway characteristics. When the width of the fairway was expanded, the ES value of $ES_L$ significantly decreased, and the ES value of $ES_A$ showed a downward trend as the ship speed increased. In addition, the overall $ES_A$ value fell from more than 10% before the expansion to less than 10% after the expansion; thus, the risk was significantly improved only by expanding the fairway width.

## 5. Conclusions and Future Work

The maritime traffic safety assessment system includes the "Harbour Approach Channels Design Guidelines" provided by the International Association of Water Transportation Facilities (PIANC) as a standard for fairway design and Korea's port and fishing port design standards. The PIANC Guide and the Port and Fishing Port Design Standards recommend designing an appropriate fairway width based on the maximum length or width of ships passing through the target sea area in the same direction. The PIANC Guide recommends setting the fairway width 3.8~18.6 times the width of the largest ship, and the port and

fishing port design standards recommend setting the fairway width at least 1~2 times the length of the largest vessel. As a result of reviewing according to this standard, it was found that the width of the target fairway was appropriate. However, as a result of performing a congestion assessment, which is an evaluation technique in the safety diagnosis system, we found that the congestion level was very high in the Yeosu fairway and the Gadeok fairway at a specific time; with the number of vessels exceeding the carrying capacity of the fairway. This presents a problem when designing a fairway based on the largest ship; and to design an appropriate fairway, the design must consider both the vessel traffic volume and the characteristics of the sea area.

To this end, in this study, a case study on the target fairway was conducted. We proposed a method for setting the appropriate width of the fairway that can solve the risk of collision between ships within a certain fairway and the risk of grounding between ships and islands, obstacles, and low water depths. The risk of collision was evaluated using the ES model, and the risk of grounding was evaluated using the IWRAP Mk II evaluation technique. If the frequency of grounding was higher than the probability of 1 in 10,000, it was judged that there was a risk of grounding. As a result of evaluating the collision risk and grounding risk for the four South Coast TSS fairways, the grounding probability was found to be lower than $10^{-4}$ for all fairways, indicating that the criteria presented in this study were satisfied. However, the overall environmental stress according to the ES model was over 10% in the Yeosu fairway, Masan fairway, and Gadeok fairway; thus, there was a risk for vessel passage. Based on these results, the width of the three fairways showing a high risk of collision was extended from 100 m to 1 km, and the evaluation was performed again using the ES model. As the stress level became less than 10%, the risk of collision was found to be low.

In the existing evaluation method, when establishing a new fairway or examining an existing fairway, the fairway width is designed based on the specifications of the largest ship. Thus, a fairway width design method was proposed that reflects the characteristics of different ships and sea areas and considers the risk of ship collision and grounding.

However, in this study, additional research is needed, due to the following limitations.

1. Diversification of evaluation models. Among the various risk assessment models, the ES model and the IWRAP Mk II model were used. The ES model quantifies the psychological risk of the operator, and the IWRAP Mk II model is used in many areas. However, because results can be derived differently, depending on various theories and research methodologies, it is necessary to derive comprehensive results through a diversification of evaluation models.

2. Ship control simulation. In the maritime traffic safety diagnosis, the fairway design and adequacy are primarily reviewed in accordance with the design standards and guidelines. In addition, ship control simulation is additionally conducted to review safety in an environment similar to the real one, as well as to examine the feasibility of restricting the fairway width according to the characteristics of the sea area. In a follow-up study, various research methods, such as ship steering simulation, will be used to improve the qualitative completeness of the research. The purpose will be to verify the safety according to the collision risk, grounding risk, and fairway width extension derived from this study, using a ship steering simulation evaluation for operators.

3. The grounding probability was evaluated for the nearest island, obstacle, and low water area, without considering the fairway lines, it is not expected that there will be any serious change in the grounding probability even if the fairway is partially expanded. However, assuming that the ship's trajectories when they navigate along the fairway follow a normal distribution, the grounding probability may increase depending on the deviation in traffic. This study evaluates the grounding probability based on actual data, it was considered too difficult to include the method of calculating the grounding probability by estimating the vessel's track within the scope of the study. Therefore, we present these notes and comments as limitations of

this study and suggest that follow-up studies deal with the estimated change in the grounding probability, due to fairway expansion, and perform verifications through actual measurements.

**Author Contributions:** Conceptualization, Y.-S.P., W.-S.K. and S.P.; methodology, W.-S.K. and M.-K.L.; validation, Y.-S.P. and W.-S.K.; writing—original draft preparation, W.-S.K.; writing—review and editing, Y.-S.P. and S.P.; visualization, W.-S.K. and M.-K.L.; supervision, Y.-S.P. All authors have read and agreed to the published version of the manuscript.

**Funding:** This research received no external funding.

**Institutional Review Board Statement:** Not applicable.

**Informed Consent Statement:** Not applicable.

**Data Availability Statement:** The data presented in this study are available on request from the corresponding author. The data are not publicly available due to privacy.

**Conflicts of Interest:** The authors declare no conflict of interest.

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
