# Peer review of "Design of Fairway Width Based on a Grounding and Collision Risk Model in the South Coast of Korean Waterways"

_applsci, doi:10.3390/app12104862_

Round 1

Reviewer 1 Report

This paper analyzed and evaluated four fairways in Korea and redesign the fairway width based on the collision and grounding probability considering vessel traffic. The environmental stress (ES) model was used to evaluate the collision risk and the IWRAP Mk II model was used to evaluate grounding frequency. The goal and results of the paper are clear to readers, but I don’t see how this can be a contribution to the methodology, which needs to be explained in detail. I have some comments on the manuscript as follows:

The authors introduced the research background and the standards implemented in fairway design. However, the necessity of research on collision risk and grounding risk is not mentioned. A few possible literature can be added, including “Personal and societal impacts of motorcycle ban policy on motorcyclists’ home-to-work morning commute in China”, “Understanding electric bike riders’ intention to violate traffic rules and accident proneness in China, and “Not all protected bike lanes are the same: Infrastructure and risk of cyclist collisions and falls leading to emergency department visits in three US cities”.

  1. Status analysis of the Fairway & 3. Conformity Review according to Fairway Design Standard

Sections 2 & 3 mainly introduced the general status, vessel traffic status, and marine traffic congestion of selected four fairways. But these sections did not discuss the meaning of collision risk and grounding risk on fairway design or clarify what the empirical did or did not do. Relevant literature should be supplemented.

  1. Re-Designing of the Appropriate Fairway Width

The ES model and the IWRAP Mk II model were used in this paper. Why did the authors use these two models? What are the advantages and disadvantages of the proposed models in comparison to currently utilized techniques? These are critical questions for readers to understand the study. Regrettably, the authors did not explain why in this section.

Reviewer 2 Report

There are a number of international recommendations for fairway design, but the recommendations by Permanent International Association of Navigation Congresses (PIANC), Puerto Del Estado (ROM 3.1) and the Ministry of Land, Infrastructure, Transport and Tourism (MLIT) are the best known. Will be interested to evaluate fairways mentioned in your paper with ROM 3.1 and MLIT too. 

Author Response

Thank you for your careful review.
The paper was revised and submitted in consideration of your comments, and answers and revision directions are provided in the attached file. Although the contents of "Comments and Suggestions for Authors
" were reviewed, the scope of the study was aimed at the southern coast of Korea, so in this paper, the port and fishing port design standards and PIANC Guide were applied in consideration of Korea's Maritime Traffic Safety Assessment Scheme.
The proposed design standards will be applied in future studies.
Thanks again for your careful review.

Please see the attachment for details.

Round 2

Reviewer 1 Report

The authors have addressed all my concerns.